# Fast Randomized Kernel Ridge Regression with Statistical Guarantees*

**Ahmed El Alaoui** †                    **Michael W. Mahoney** ‡
† Electrical Engineering and Computer Sciences
‡ Statistics and International Computer Science Institute
University of California, Berkeley, Berkeley, CA 94720.
{elalaoui@eecs,mmahoney@stat}.berkeley.edu

## Abstract

One approach to improving the running time of kernel-based methods is to build a small sketch of the kernel matrix and use it in lieu of the full matrix in the machine learning task of interest. Here, we describe a version of this approach that comes with running time guarantees as well as improved guarantees on its statistical performance. By extending the notion of *statistical leverage scores* to the setting of kernel ridge regression, we are able to identify a sampling distribution that reduces the size of the sketch (i.e., the required number of columns to be sampled) to the *effective dimensionality* of the problem. This latter quantity is often much smaller than previous bounds that depend on the *maximal degrees of freedom*. We give an empirical evidence supporting this fact. Our second contribution is to present a fast algorithm to quickly compute coarse approximations to these scores in time linear in the number of samples. More precisely, the running time of the algorithm is $O(np^2)$ with $p$ only depending on the trace of the kernel matrix and the regularization parameter. This is obtained via a variant of squared length sampling that we adapt to the kernel setting. Lastly, we discuss how this new notion of the leverage of a data point captures a fine notion of the difficulty of the learning problem.

## 1   Introduction

We consider the low-rank approximation of symmetric positive semi-definite (SPSD) matrices that arise in machine learning and data analysis, with an emphasis on obtaining good statistical guarantees. This is of interest primarily in connection with kernel-based machine learning methods. Recent work in this area has focused on one or the other of two very different perspectives: an *algorithmic perspective*, where the focus is on running time issues and worst-case quality-of-approximation guarantees, given a fixed input matrix; and a *statistical perspective*, where the goal is to obtain good inferential properties, under some hypothesized model, by using the low-rank approximation in place of the full kernel matrix. The recent results of Gittens and Mahoney [2] provide the strongest example of the former, and the recent results of Bach [3] are an excellent example of the latter. In this paper, we combine ideas from these two lines of work in order to obtain a fast randomized kernel method with statistical guarantees that are improved relative to the state-of-the-art.

To understand our approach, recall that several papers have established the crucial importance—from the algorithmic perspective—of the *statistical leverage scores*, as they capture structural non-uniformities of the input matrix and they can be used to obtain very sharp worst-case approximation guarantees. See, e.g., work on CUR matrix decompositions [5, 6], work on the the fast approximation of the statistical leverage scores [7], and the recent review [8] for more details. Here, we

simply note that, when restricted to an $n \times n$ SPSD matrix $K$ and a rank parameter $k$, the statistical leverage scores relative to the best rank-$k$ approximation to $K$, call them $\ell_i$, for $i \in \{1, \ldots, n\}$, are the diagonal elements of the projection matrix onto the best rank-$k$ approximation of $K$. That is, $\ell_i = \text{diag}(K_k K_k^\dagger)_i$, where $K_k$ is the best rank $k$ approximation of $K$ and where $K_k^\dagger$ is the Moore-Penrose inverse of $K_k$. The recent work by Gittens and Mahoney [2] showed that qualitatively improved worst-case bounds for the low-rank approximation of SPSD matrices could be obtained in one of two related ways: either compute (with the fast algorithm of [7]) approximations to the leverage scores, and use those approximations as an importance sampling distribution in a random sampling algorithm; or rotate (with a Gaussian-based or Hadamard-based random projection) to a random basis where those scores are uniformized, and sample randomly in that rotated basis.

In this paper, we extend these ideas, and we show that—from the statistical perspective—we are able to obtain a low-rank approximation that comes with improved statistical guarantees by using a variant of this more traditional notion of statistical leverage. In particular, we improve the recent bounds of Bach [3], which provides the first known statistical convergence result when substituting the kernel matrix by its low-rank approximation. To understand the connection, recall that a key component of Bach's approach is the quantity $d_{\text{mof}} = n\|\text{diag}(K(K + n\lambda I)^{-1})\|_\infty$, which he calls the *maximal marginal degrees of freedom*.[1] Bach's main result is that by constructing a low-rank approximation of the original kernel matrix by sampling uniformly at random $p = O(d_{\text{mof}}/\epsilon)$ columns, i.e., performing the vanilla Nyström method, and then by using this low-rank approximation in a prediction task, the statistical performance is within a factor of $1 + \epsilon$ of the performance when the entire kernel matrix is used. Here, we show that this uniform sampling is suboptimal. We do so by sampling with respect to a coarse but quickly-computable approximation of a variant to the statistical leverage scores, given in Definition 1 below, and we show that we can obtain similar $1 + \epsilon$ guarantees by sampling only $O(d_{\text{eff}}/\epsilon)$ columns, where $d_{\text{eff}} = \text{Tr}(K(K + n\lambda I)^{-1}) < d_{\text{mof}}$. The quantity $d_{\text{eff}}$ is called the *effective dimensionality* of the learning problem, and it can be interpreted as the implicit number of parameters in this nonparametric setting [9, 10].

We expect that our results and insights will be useful much more generally. As an example of this, we can directly compare the Nyström sampling method to a related divide-and-conquer approach, thereby answering an open problem of Zhang et al. [9]. Recall that the Zhang et al. divide-and-conquer method consists of dividing the dataset $\{(x_i, y_i)\}_{i=1}^n$ into $m$ random partitions of equal size, computing estimators on each partition in parallel, and then averaging the estimators. They prove the minimax optimality of their estimator, although their multiplicative constants are suboptimal; and, in terms of the number of kernel evaluations, their method requires $m \times (n/m)^2$, with $m$ in the order of $n/d_{\text{eff}}^2$, which gives a total number of $O(nd_{\text{eff}}^2)$ evaluations. They noticed that the scaling of their estimator was *not* directly comparable to that of the Nyström sampling method (which was proven to only require $O(nd_{\text{mof}})$ evaluations, if the sampling is uniform [3]), and they left it as an open problem to determine which if either method is fundamentally better than the other. Using our Theorem 3, we are able to put both results on a common ground for comparison. Indeed, the estimator obtained by our *non-uniform* Nyström sampling requires only $O(nd_{\text{eff}})$ kernel evaluations (compared to $O(nd_{\text{eff}}^2)$ and $O(nd_{\text{mof}})$), and it obtains the same bound on the statistical predictive performance as in [3]. In this sense, our result combines "the best of both worlds," by having the reduced sample complexity of [9] and the sharp approximation bound of [3].

## 2 Preliminaries and notation

Let $\{(x_i, y_i)\}_{i=1}^n$ be $n$ pairs of points in $\mathcal{X} \times \mathcal{Y}$, where $\mathcal{X}$ is the input space and $\mathcal{Y}$ is the response space. The kernel-based learning problem can be cast as the following minimization problem:

$$\min_{f \in \mathcal{F}} \frac{1}{n} \sum_{i=1}^n \ell(y_i, f(x_i)) + \frac{\lambda}{2}\|f\|_{\mathcal{F}}^2, \tag{1}$$

where $\mathcal{F}$ is a reproducing kernel Hilbert space and $\ell : \mathcal{Y} \times \mathcal{Y} \to \mathbb{R}$ is a loss function. We denote by $k : \mathcal{X} \times \mathcal{X} \to \mathbb{R}$ the positive definite kernel corresponding to $\mathcal{F}$ and by $\phi : \mathcal{X} \to \mathcal{F}$ a corresponding feature map. That is, $k(x, x') = \langle \phi(x), \phi(x') \rangle_{\mathcal{F}}$ for every $x, x' \in \mathcal{X}$. The representer theorem [11, 12] allows us to reduce Problem (1) to a finite-dimensional optimization problem, in which

case Problem (1) boils down to finding the vector $\alpha \in \mathbb{R}^n$ that solves

$$\min_{\alpha \in \mathbb{R}^n} \frac{1}{n} \sum_{i=1}^{n} \ell(y_i, (K\alpha)_i) + \frac{\lambda}{2} \alpha^\top K\alpha, \tag{2}$$

where $K_{ij} = k(x_i, x_j)$. We let $U\Sigma U^\top$ be the eigenvalue decomposition of $K$, with $\Sigma = \text{Diag}(\sigma_1, \cdots, \sigma_n)$, $\sigma_1 \geq \cdots \geq \sigma_n \geq 0$, and $U$ an orthogonal matrix. The underlying data model is

$$y_i = f^*(x_i) + \sigma^2 \xi_i \quad i = 1, \cdots, n$$

with $f^* \in \mathcal{F}$, $(x_i)_{1 \leq i \leq n}$ a deterministic sequence and $\xi_i$ are i.i.d. standard normal random variables. We consider $\ell$ to be the squared loss, in which case we will be interested in the mean squared error as a measure of statistical risk: for any estimator $\hat{f}$, let

$$\mathcal{R}(\hat{f}) := \frac{1}{n} \mathsf{E}_\xi \|\hat{f} - f^*\|_2^2 \tag{3}$$

be the risk function of $\hat{f}$ where $\mathsf{E}_\xi$ denotes the expectation under the randomness induced by $\xi$. In this setting the problem is called *Kernel Ridge Regression* (KRR). The solution to Problem (2) is $\alpha = (K + n\lambda I)^{-1} y$, and the estimate of $f^*$ at any training point $x_i$ is given by $\hat{f}(x_i) = (K\alpha)_i$. We will use $\hat{f}_K$ as a shorthand for the vector $(\hat{f}(x_i))_{1 \leq i \leq n} \in \mathbb{R}^n$ when the matrix $K$ is used as a kernel matrix. This notation will be used accordingly for other kernel matrices (e.g. $\hat{f}_L$ for a matrix $L$). Recall that the risk of the estimator $\hat{f}_K$ can then be decomposed into a bias and variance term:

$$\mathcal{R}(\hat{f}_K) = \frac{1}{n} \mathsf{E}_\xi \|K(K + n\lambda I)^{-1}(f^* + \sigma^2 \xi) - f^*\|_2^2$$

$$= \frac{1}{n} \|(K(K + n\lambda I)^{-1} - I) f^*\|_2^2 + \frac{\sigma^2}{n} \mathsf{E}_\xi \|K(K + n\lambda I)^{-1} \xi\|_2^2$$

$$= n\lambda^2 \|(K + n\lambda I)^{-1} f^*\|_2^2 + \frac{\sigma^2}{n} \mathsf{Tr}(K^2(K + n\lambda I)^{-2})$$

$$:= \quad \text{bias}(K)^2 \quad + \quad \text{variance}(K). \tag{4}$$

Solving Problem (2), either by a direct method or by an optimization algorithm needs at least a quadratic and often cubic running time in $n$ which is prohibitive in the large scale setting. The so-called Nyträm method approximates the solution to Problem (2) by substituting $K$ with a low-rank approximation to $K$. In practice, this approximation is often not only fast to construct, but the resulting learning problem is also often easier to solve [13, 14, 15, 2]. The method operates as follows. A small number of columns $K_1, \cdots, K_p$ are randomly sampled from $K$. If we let $C = [K_1, \cdots, K_p] \in \mathbb{R}^{n \times p}$ denote the matrix containing the sampled columns, $W \in \mathbb{R}^{p \times p}$ the overlap between $C$ and $C^\top$ in $K$, then the Nyström approximation of $K$ is the matrix

$$L = CW^\dagger C^\top.$$

More generally, if we let $S \in R^{n \times p}$ be an arbitrary *sketching matrix*, i.e., a tall and skinny matrix that, when left-multiplied by $K$, produces a "sketch" of $K$ that preserves some desirable properties, then the Nyström approximation associated with $S$ is

$$L = KS(S^\top KS)^\dagger S^\top K.$$

For instance, for random sampling algorithms, $S$ would contain a non-zero entry at position $(i, j)$ if the $i$-th column of $K$ is chosen at the $j$-th trial of the sampling process. Alternatively, $S$ could also be a random projection matrix; or $S$ could be constructed with some other (perhaps deterministic) method, as long as it verifies some structural properties, depending on the application [8, 2, 6, 5].

We will focus in this paper on analyzing this approximation in the statistical prediction context related to the estimation of $f^*$ by solving Problem (2). We proceed by revisiting and improving upon prior results from three different areas. The first result (Theorem 1) is on the behavior of the bias of $\hat{f}_L$, when $L$ is constructed using a general sketching matrix $S$. This result underlies the statistical analysis of the Nyström method. To see this, first, it is not hard to prove that $L \preceq K$ in the sense of usual the order on the positive semi-definite cone. Second, one can prove that the variance is matrix-increasing, hence the variance will decrease when replacing $K$ by $L$. On the other

hand, the bias (while *not* matrix monotone in general) can be proven to not increase too much when replacing $K$ by $L$. This latter statement will be the main technical difficulty for obtaining a bound on $\mathcal{R}(\hat{f}_L)$ (see Appendix A). A form of this result is due to Bach [3] in the case where $S$ is a uniform sampling matrix. The second result (Theorem 2) is a concentration bound for approximating matrix multiplication when the rank-one components of the product are sampled non uniformly. This result is derived from the matrix Bernstein inequality, and yields a sharp quantification of the deviation of the approximation from the true product. The third result (Definition 1) is an extension of the definition of the leverage scores to the context of kernel ridge regression. Whereas the notion of leverage is established as an algorithmic tool in randomized linear algebra, we introduce a natural counterpart of it to this statistical setting. By combining these contributions, we are able to give a sharp statistical statement on the behavior of the Nyström method if one is allowed to sample non uniformly. All the proofs are deferred to the appendix (or see [1]).

## 3 Revisiting prior work and new results

### 3.1 A structural result

We begin by stating a "structural" result that upper-bounds the bias of the estimator constructed using the approximation $L$. This result is deterministic: it only depends on the properties of the input data, and holds for *any* sketching matrix $S$ that satisfies certain conditions. This way the randomness of the construction of $S$ is decoupled from the rest of the analysis. We highlight the fact that this view offers a possible way of improving the current results since a better construction of $S$ -whether deterministic or random- satisfying the data-related conditions would immediately lead to down stream algorithmic and statistical improvements in this setting.

**Theorem 1.** *Let $S \in \mathbb{R}^{n \times p}$ be a sketching matrix and $L$ the corresponding Nyström approximation. For $\gamma > 0$, let $\Phi = \Sigma(\Sigma + n\gamma I)^{-1}$. If the sketching matrix $S$ satisfies $\lambda_{\max}\left(\Phi - \Phi^{1/2} U^\top S S^\top U \Phi^{1/2}\right) \leq t$ for $t \in (0,1)$ and $\lambda \geq \frac{1}{1-t}\|S\|_{op}^2 \cdot \frac{\lambda_{\max}(K)}{n}$, where $\lambda_{\max}$ denotes the maximum eigenvalue and $\|\cdot\|_{op}$ is the operator norm then*

$$bias(L) \leq \left(1 + \frac{\gamma/\lambda}{1-t}\right) bias(K). \tag{5}$$

In the special case where $S$ contains one non zero entry equal to $1/\sqrt{pn}$ in every column with $p$ the number of sampled columns, the result and its proof can be found in [3] (appendix B.2), although we believe that their argument contains a problematic statement. We propose an alternative and complete proof in Appendix A. The subsequent analysis unfolds in two steps: (1) assuming the sketching matrix $S$ satisfies the conditions stated in Theorem 1, we will have $\mathcal{R}(\hat{f}_L) \lesssim \mathcal{R}(\hat{f}_K)$, and (2) matrix concentration is used to show that an appropriate random construction of $S$ satisfies the said conditions. We start by stating the concentration result that is the source of our improvement (section 3.2), define a notion of statistical leverage scores (section 3.3), and then state and prove the main statistical result (Theorem 3 section 3.4). We then present our main algorithmic result consisting of a fast approximation to this new notion of leverage scores (section 3.5).

### 3.2 A concentration bound on matrix multiplication

Next, we state our result for approximating matrix products of the form $\Psi\Psi^\top$ when a few columns from $\Psi$ are sampled to form the approximate product $\Psi_I \Psi_I^\top$ where $\Psi_I$ contains the chosen columns. The proof relies on a matrix Bernstein inequality (see e.g. [16]) and is presented at the end of the paper (Appendix B).

**Theorem 2.** *Let $n, m$ be positive integers. Consider a matrix $\Psi \in \mathbb{R}^{n \times m}$ and denote by $\psi_i$ the $i^{th}$ column of $\Psi$. Let $p \leq m$ and $I = \{i_1, \cdots, i_p\}$ be a subset of $\{1, \cdots, m\}$ formed by $p$ elements chosen randomly with replacement, according to the distribution*

$$\forall i \in \{1, \cdots, m\} \quad \mathsf{Pr}(choosing\ i) = p_i \geq \beta \frac{\|\psi_i\|_2^2}{\|\Psi\|_F^2} \tag{6}$$

*for some $\beta \in (0, 1]$. Let $S \in \mathbb{R}^{n \times p}$ be a sketching matrix such that $S_{ij} = 1/\sqrt{p \cdot p_{i_j}}$ only if $i = i_j$ and 0 elsewhere. Then*

$$\Pr\left(\lambda_{\max}\left(\Psi\Psi^\top - \Psi SS^\top\Psi^\top\right) \geq t\right) \leq n \exp\left(\frac{-pt^2/2}{\lambda_{\max}(\Psi\Psi^\top)(\|\Psi\|_F^2/\beta + t/3)}\right). \tag{7}$$

**Remarks:** 1. This result will be used for $\Psi = \Phi^{1/2}U^\top$, in conjunction with Theorem 1 to prove our main result in Theorem 3. Notice that $\Psi^\top$ is a scaled version of the eigenvectors, with a scaling given by the diagonal matrix $\Phi = \Sigma(\Sigma + n\gamma I)^{-1}$ which should be considered as "soft projection" matrix that smoothly selects the top part of the spectrum of $K$. The setting of Gittens et al. [2], in which $\Phi$ is a 0-1 diagonal is the closest analog of our setting.

2. It is known that $p_i = \frac{\|\psi_i\|_2^2}{\|\Psi\|_F^2}$ is the optimal sampling distribution in terms of minimizing the expected error $\mathsf{E}\|\Psi\Psi^\top - \Psi SS^\top\Psi^\top\|_F^2$ [17]. The above result exhibits a robustness property by allowing the chosen sampling distribution to be different from the optimal one by a factor $\beta$.[2] The sub-optimality of such a distribution is reflected in the upper bound (7) by the amplification of the squared Frobenius norm of $\Psi$ by a factor $1/\beta$. For instance, if the sampling distribution is chosen to be uniform, i.e. $p_i = 1/m$, then the value of $\beta$ for which (6) is tight is $\frac{\|\Psi\|_F^2}{m \max_i \|\psi_i\|_2^2}$, in which case we recover a concentration result proven by Bach [3]. Note that Theorem 2 is derived from one of the state-of-the-art bounds on matrix concentration, but it is one among many others in the literature; and while it constitutes the base of our improvement, it is possible that a concentration bound more tailored to the problem might yield sharper results.

### 3.3 An extended definition of leverage

We introduce an extended notion of leverage scores that is specifically tailored to the ridge regression problem, and that we call the $\lambda$-*ridge leverage scores*.

**Definition 1.** *For $\lambda > 0$, the $\lambda$-ridge leverage scores associated with the kernel matrix $K$ and the parameter $\lambda$ are*

$$\forall i \in \{1, \cdots, n\}, \qquad l_i(\lambda) = \sum_{j=1}^n \frac{\sigma_j}{\sigma_j + n\lambda}U_{ij}^2. \tag{8}$$

Note that $l_i(\lambda)$ is the $i^{th}$ diagonal entry of $K(K + n\lambda I)^{-1}$. The quantities $(l_i(\lambda))_{1 \leq i \leq n}$ are in this setting the analogs of the so-called *leverage scores* in the statistical literature, as they characterize the data points that "stick out", and consequently that most affect the result of a statistical procedure. They are classically defined as the row norms of the left singular matrix $U$ of the input matrix, and they have been used in regression diagnostics for outlier detection [18], and more recently in randomized matrix algorithms as they often provide an optimal importance sampling distribution for constructing random sketches for low rank approximation [17, 19, 5, 6, 2] and least squares regression [20] when the input matrix is tall and skinny ($n \geq m$). In the case where the input matrix is square, this definition is vacuous as the row norms of $U$ are all equal to 1. Recently, Gittens and Mahoney [2] used a truncated version of these scores (that they called *leverage scores relative to the best rank-$k$ space*) to obtain the best algorithmic results known to date on low rank approximation of positive semi-definite matrices. Definition 1 is a weighted version of the classical leverage scores, where the weights depend on the spectrum of $K$ and a regularization parameter $\lambda$. In this sense, it is an interpolation between Gittens' scores and the classical (tall-and-skinny) leverage scores, where the parameter $\lambda$ plays the role of a rank parameter. In addition, we point out that Bach's maximal degrees of freedom $d_{\mathrm{mof}}$ is to the $\lambda$-ridge leverage scores what the *coherence* is to Gittens' leverage scores, i.e. their (scaled) maximum value: $d_{\mathrm{mof}}/n = \max_i l_i(\lambda)$; and that while the sum of Gittens' scores is the rank parameter $k$, the sum of the $\lambda$-ridge leverage scores is the effective dimensionality $d_{\mathrm{eff}}$. We argue in the following that Definition 1 provides a relevant notion of leverage in the context of kernel ridge regression. It is the natural counterpart of the algorithmic notion of leverage in the prediction context. We use it in the next section to make a statistical statement on the performance of the Nyström method.

## 3.4  Main statistical result: an error bound on approximate kernel ridge regression

Now we are able to give an improved version of a theorem by Bach [3] that establishes a performance guaranty on the use of the Nyström method in the context of kernel ridge regression. It is improved in the sense that the sufficient number of columns that should be sampled in order to incur no (or little) loss in the prediction performance is lower. This is due to a more data-sensitive way of sampling the columns of $K$ (depending on the $\lambda$-ridge leverage scores) during the construction of the approximation $L$. The proof is in Appendix C.

**Theorem 3.** *Let $\lambda, \epsilon > 0$, $\rho \in (0, 1/2)$, $n \geq 2$ and L be a Nyström approximation of K by choosing p columns randomly with replacement according to a probability distribution $(p_i)_{1 \leq i \leq n}$ such that $\forall i \in \{1, \cdots, n\}, \quad p_i \geq \beta \cdot l_i(\lambda\epsilon)/\sum_{i=1}^n l_i(\lambda\epsilon)$ for some $\beta \in (0,1]$. Let $\underline{l} \leq \min_i l_i(\lambda\epsilon)$. If*

$$p \geq 8 \left( \frac{d_{\textit{eff}}}{\beta} + \frac{1}{6} \right) \log \left( \frac{n}{\rho} \right) \; \textit{ and } \; \lambda \geq 2 \left( 1 + \frac{1}{\underline{l}} \right) \frac{\lambda_{\max}(K)}{n},$$

*with $d_{\textit{eff}} = \sum_{i=1}^n l_i(\lambda\epsilon) = \mathsf{Tr}(K(K + n\lambda\epsilon I)^{-1})$ then*

$$\mathcal{R}(\hat{f}_L) \leq (1 + 2\epsilon)^2 \mathcal{R}(\hat{f}_K)$$

*with probability at least $1 - 2\rho$, where $(l_i)_i$ are introduced in Definition 1 and $\mathcal{R}$ is defined in* (3).

Theorem 3 asserts that substituting the kernel matrix $K$ by a Nyström approximation of rank $p$ in the KRR problem induces an arbitrarily small prediction loss, provided that $p$ scales linearly with the effective dimensionality $d_{\text{eff}}$[3] and that $\lambda$ is not too small[4]. The leverage-based sampling appears to be crucial for obtaining this dependence, as the $\lambda$-ridge leverage scores provide information on which columns -and hence which data points- capture most of the difficulty of the estimation problem. Also, as a sanity check, the smaller the target accuracy $\epsilon$, the higher $d_{\text{eff}}$, and the more uniform the sampling distribution $(l_i(\lambda\epsilon))_i$ becomes. In the limit $\epsilon \to 0$, $p$ is in the order of $n$ and the scores are uniform, and the method is essentially equivalent to using the entire matrix $K$. Moreover, if the sampling distribution $(p_i)_i$ is a factor $\beta$ away from optimal, a slight oversampling (i.e. increase $p$ by $1/\beta$) achieves the same performance. In this sense, the above result shows robustness to the sampling distribution. This property is very beneficial from an implementation point of view, as the error bounds still hold when only an approximation of the leverage scores is available. If the columns are sampled uniformly, a worse lower bound on $p$ that depends on $d_{\text{mof}}$ is obtained [3].

## 3.5  Main algorithmic result: a fast approximation to the $\lambda$-ridge leverage scores

Although the $\lambda$-ridge leverage scores can be naively computed using SVD, the exact computation is as costly as solving the original Problem (2). Therefore, the central role they play in the above result motivates the problem of a fast approximation, in a similar way the importance of the usual leverage scores has motivated Drineas et al. to approximate them is random projection time [7]. A success in this task will allow us to combine the running time benefits with the improved statistical guarantees we have provided.

**Algorithm:**

- **Inputs**: data points $(x_i)_{1 \leq i \leq n}$, probability vector $(p_i)_{1 \leq i \leq n}$, sampling parameter $p \in \{1, 2, \cdots\}$, $\lambda > 0$, $\epsilon \in (0, 1/2)$.
- **Output**: $(\tilde{l}_i)_{1 \leq i \leq n}$ $\epsilon$-approximations to $(l_i(\lambda))_{1 \leq i \leq n}$.
1. Sample $p$ data points from $(x_i)_{1 \leq i \leq n}$ with replacement with probabilities $(p_i)_{1 \leq i \leq n}$.
2. Compute the corresponding columns $K_1, \cdots, K_p$ of the kernel matrix.
3. Construct $C = [K_1, \cdots, K_p] \in \mathbb{R}^{n \times p}$ and $W \in \mathbb{R}^{p \times p}$ as presented in Section 2.
4. Construct $B \in \mathbb{R}^{n \times p}$ such that $BB^\top = CW^\dagger C^\top$.
5. For every $i \in \{1, \cdots, n\}$, set

$$\tilde{l}_i = B_i^\top (B^\top B + n\lambda I)^{-1} B_i \tag{9}$$

where $B_i$ is the $i$-th row of $B$, and return it.

**Running time:** The running time of the above algorithm is dominated by steps 4 and 5. Indeed, constructing $B$ can be done using a Cholesky factorization on $W$ and then a multiplication of $C$ by the inverse of the obtained Cholesky factor, which yields a running time of $O(p^3 + np^2)$. Computing the approximate leverage scores $(\tilde{l}_i)_{1 \leq i \leq n}$ in step 5 also runs in $O(p^3 + np^2)$. Thus, for $p \ll n$, the overall algorithm runs in $O(np^2)$. Note that formula (9) only involves matrices and vectors of size $p$ (everything is computed in the smaller dimension $p$), and the fact that this yields a correct approximation relies on the matrix inversion lemma (see proof in Appendix D). Also, only the relevant columns of $K$ are computed and we never have to form the entire kernel matrix. This improves over earlier models [2] that require that all of $K$ has to be written down in memory. The improved running time is obtained by considering the construction (9) which is quite different from the regular setting of approximating the leverage scores of a rectangular matrix [7]. We now give both additive and multiplicative error bounds on its approximation quality.

**Theorem 4.** *Let $\epsilon \in (0, 1/2)$, $\rho \in (0, 1)$ and $\lambda > 0$. Let $L$ be a Nyström approximation of $K$ by choosing $p$ columns at random with probabilities $p_i = K_{ii}/\text{Tr}(K)$, $i = 1, \cdots, n$. If*

$$p \geq 8 \left( \frac{\text{Tr}(K)}{n\lambda\epsilon} + \frac{1}{6} \right) \log \left( \frac{n}{\rho} \right)$$

*then we have $\forall i \in \{1, \cdots, n\}$*

$$\text{(additive error bound)} \quad l_i(\lambda) - 2\epsilon \leq \tilde{l}_i \leq l_i(\lambda)$$

*and*

$$\text{(multiplicative error bound)} \quad \left( \frac{\sigma_n - n\lambda\epsilon}{\sigma_n + n\lambda\epsilon} \right) l_i(\lambda) \leq \tilde{l}_i \leq l_i(\lambda)$$

*with probability at least $1 - \rho$.*

**Remarks:** 1. Theorem 4 states that if the columns of $K$ are sampled proportionally to $K_{ii}$ then $O(\frac{\text{Tr}(K)}{n\lambda})$ is a sufficient number of samples. Recall that $K_{ii} = \|\phi(x_i)\|_{\mathcal{F}}^2$, so our procedure is akin to sampling according to the squared lengths of the data vectors, which has been extensively used in different contexts of randomized matrix approximation [21, 17, 19, 8, 2].

2. Due to how $\lambda$ is defined in eq. (1) the $n$ in the denominator is artificial: $n\lambda$ should be thought of as a "rescaled" regularization parameter $\lambda'$. In some settings, the $\lambda$ that yields the best generalization error scales like $O(1/\sqrt{n})$, hence $p = O(\text{Tr}(K)/\sqrt{n})$ is sufficient. On the other hand, if the columns are sampled uniformly, one would get $p = O(d_{\text{mof}}) = O(n \max_i l_i(\lambda))$.

## 4 Experiments

We test our results based on several datasets: one synthetic regression problem from [3] to illustrate the importance of the $\lambda$-ridge leverage scores, the *Pumadyn* family consisting of three datasets *pumadyn-32fm*, *pumadyn-32fh* and *pumadyn-32nh* [5] and the *Gas Sensor Array Drift Dataset* from the UCI database[6]. The synthetic case consists of a regression problem on the interval $\mathcal{X} = [0, 1]$ where, given a sequence $(x_i)_{1 \leq i \leq n}$ and a sequence of noise $(\epsilon_i)_{1 \leq i \leq n}$, we observe the sequence

$$y_i = f(x_i) + \sigma^2 \epsilon_i, \quad i \in \{1, \cdots, n\}.$$

The function $f$ belongs to the RKHS $\mathcal{F}$ generated by the kernel $k(x, y) = \frac{1}{(2\beta)!} B_{2\beta}(x - y - \lfloor x - y \rfloor)$ where $B_{2\beta}$ is the $2\beta$-th Bernoulli polynomial [3]. One important feature of this regression problem is the distribution of the points $(x_i)_{1 \leq i \leq n}$ on the interval $\mathcal{X}$: if they are spread uniformly over the interval, the $\lambda$-ridge leverage scores $(\tilde{l}_i(\lambda))_{1 \leq i \leq n}$ are uniform for every $\lambda > 0$, and uniform column sampling is optimal in this case. In fact, if $x_i = \frac{i-1}{n}$ for $i = 1, \cdots, n$, the kernel matrix $K$ is a circulant matrix [3], in which case, we can prove that the $\lambda$-ridge leverage scores are constant. Otherwise, if the data points are distributed asymmetrically on the interval, the $\lambda$-ridge leverage scores are non uniform, and importance sampling is beneficial (Figure 1). In this experiment, the data points $x_i \in (0, 1)$ have been generated with a distribution symmetric about $\frac{1}{2}$, having a high density on the borders of the interval $(0, 1)$ and a low density on the center of the interval. The number of observations is $n = 500$. On Figure 1, we can see that there are few data points with

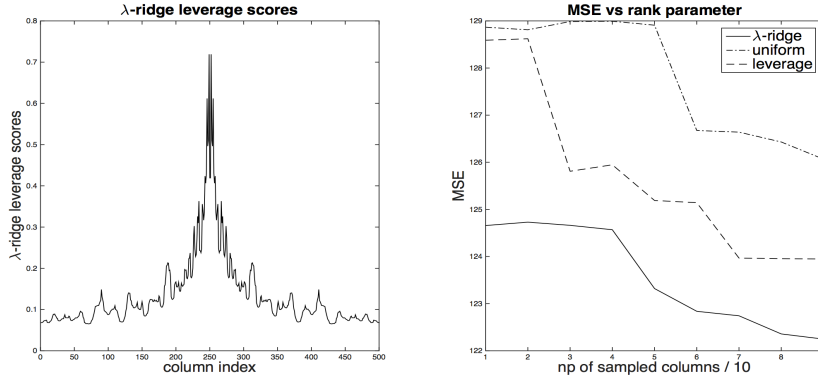

Figure 1: The $\lambda$-ridge leverage scores for the synthetic Bernoulli data set described in the text (left) and the MSE risk vs. the number of sampled columns used to construct the Nyström approximation for different sampling methods (right).

high leverage, and those correspond to the region that is underrepresented in the data sample (i.e. the region close to the center of the interval since it is the one that has the lowest density of observations). The $\lambda$-ridge leverage scores are able to capture the importance of these data points, thus providing a way to detect them (e.g. with an analysis of outliers), had we not known their existence.

For all datasets, we determine $\lambda$ and the band width of $k$ by cross validation, and we compute the effective dimensionality $d_{\text{eff}}$ and the maximal degrees of freedom $d_{\text{mof}}$. Table 1 summarizes the experiments. It is often the case that $d_{\text{eff}} \ll d_{\text{mof}}$ and $\mathcal{R}(\hat{f}_L)/\mathcal{R}(\hat{f}_K) \simeq 1$, in agreement with Theorem 3.

| kernel | dataset | $n$ | nb. feat | band width | $\lambda$ | $d_{\text{eff}}$ | $d_{\text{mof}}$ | risk ratio $\mathcal{R}(\hat{f}_L)/\mathcal{R}(\hat{f}_K)$ | |
|---|---|---|---|---|---|---|---|---|---|
| Bern | Synth | 500 | - | - | $1e{-}6$ | 24 | 500 | 1.01 | $(p = 2d_{\text{eff}})$ |
| Linear | Gas2 | 1244 | 128 | - | $1e{-}3$ | 126 | 1244 | 1.10 | $(p = 2d_{\text{eff}})$ |
| | Gas3 | 1586 | 128 | - | $1e{-}3$ | 125 | 1586 | 1.09 | $(p = 2d_{\text{eff}})$ |
| | Pum-32fm | 2000 | 32 | - | $1e{-}3$ | 31 | 2000 | 0.99 | $(p = 2d_{\text{eff}})$ |
| | Pum-32fh | 2000 | 32 | - | $1e{-}3$ | 31 | 2000 | 0.99 | $(p = 2d_{\text{eff}})$ |
| | Pum-32nh | 2000 | 32 | - | $1e{-}3$ | 32 | 2000 | 0.99 | $(p = 2d_{\text{eff}})$ |
| RBF | Gas2 | 1244 | - | 1 | $4.5e{-}4$ | 1135 | 1244 | 1.56 | $(p = d_{\text{eff}})$ |
| | Gas3 | 1586 | - | 1 | $5e{-}4$ | 1450 | 1586 | 1.50 | $(p = d_{\text{eff}})$ |
| | Pum-32fm | 2000 | - | 5 | 0.5 | 142 | 1897 | 1.00 | $(p = d_{\text{eff}})$ |
| | Pum-32fh | 2000 | - | 5 | $5e{-}2$ | 747 | 1989 | 1.00 | $(p = d_{\text{eff}})$ |
| | Pum-32nh | 2000 | - | 5 | $1.3e{-}2$ | 1337 | 1997 | 0.99 | $(p = d_{\text{eff}})$ |

Table 1: Parameters and quantities of interest for the different datasets and using different kernels: the synthetic dataset using the Bernoulli kernel (denoted by Synth), the Gas Sensor Array Drift Dataset (batches 2 and 3, denoted by Gas2 and Gas3) and the Pumadyn datasets (Pum-32fm, Pum-32fh, Pum-32nh) using linear and RBF kernels.

## 5 Conclusion

We showed in this paper that in the case of kernel ridge regression, the sampling complexity of the Nyström method can be reduced to the effective dimensionality of the problem, hence bridging and improving upon different previous attempts that established weaker forms of this result. This was achieved by defining a natural analog to the notion of leverage scores in this statistical context, and using it as a column sampling distribution. We obtained this result by combining and improving upon results that have emerged from two different perspectives on low rank matrix approximation. We also present a way to approximate these scores that is computationally tractable, i.e. runs in time $O(np^2)$ with $p$ depending only on the trace of the kernel matrix and the regularization parameter. One natural unanswered question is whether it is possible to further reduce the sampling complexity, or is the effective dimensionality also a lower bound on $p$? And as pointed out by previous work [22, 3], it is likely that the same results hold for smooth losses beyond the squared loss (e.g. logistic regression). However the situation is unclear for non-smooth losses (e.g. support vector regression).

**Acknowledgements:** We thank Xixian Chen for pointing out a mistake in an earlier draft of this paper [1]. We thank Francis Bach for stimulating discussions and for contributing to a rectified proof of Theorem 1. We thank Jason Lee and Aaditya Ramdas for fruitful discussions regarding the proof of Theorem 1. We thank Yuchen Zhang for pointing out the connection to his work.

## Footnotes

*A technical report version of this conference paper is available at [1].

[1]We will refer to it as the maximal degrees of freedom.

[2]In their work [17], Drineas et al. have a comparable robust statement for controlling the expected error. Our result is a robust quantification of the tail probability of the error, which is a much stronger statement.

[3]Note that $d_{\text{eff}}$ depends on the precision parameter $\epsilon$, which is absent in the classical definition of the effective dimensionality [10, 9, 3] However, the following bound holds: $d_{\text{eff}} \leq \frac{1}{\epsilon}\mathsf{Tr}(K(K + n\lambda I)^{-1})$.

[4]This condition on $\lambda$ is not necessary if one constructs $L$ as $KS(S^\top KS + n\lambda\epsilon I)^{-1}S^\top K$ (see proof).

[5] http://www.cs.toronto.edu/~delve/data/pumadyn/desc.html

[6] https://archive.ics.uci.edu/ml/datasets/Gas+Sensor+Array+Drift+Dataset

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
