[Supplementary Material]

# Supplementary material

## A Proof of Theorem 1

**Note:** This proof is inspired by one of Bach [1]. We extend their result to the case of a general sketching matrix $S$. Moreover, we believe their argument contains two problematic statements (about monotonicity of the bias) that we rectify with Lemma 2 and Lemma 3 below. Their result therefore holds also true with minimal change based on this argument.

For kernel ridge regression, the bias of the estimator $\hat{f}_K$ can be expressed as

$$\text{bias}(K)^2 = n\lambda^2 \|(K + n\lambda I)^{-1} f^*\|^2$$
$$= n\lambda^2 {f^*}^\top (K + n\lambda I)^{-2} f^*.$$

For $\gamma > 0$, we consider again the regularized approximation $L_\gamma = KS(S^\top KS + n\gamma I)^{-1}S^\top K$ with $S \in \mathbb{R}^{n \times p}$ the sketching matrix. The result of the theorem follows from the three following lemmas.

**Lemma 1.** *Let* $K = U\Sigma U^\top$ *where* $U$ *is orthogonal and* $\Sigma$ *diagonal positive. We have*

$$L_\gamma \preceq L \preceq K. \tag{1}$$

*Moreover, let*

$$D = \Phi - \Phi^{1/2} U^\top S S^\top U \Phi^{1/2}$$

*with* $\Phi = \Sigma(\Sigma + n\gamma I)^{-1}$. *If* $\lambda_{\max}(D) \leq t$ *for* $t \in (0,1)$ *then*

$$0 \preceq K - L_\gamma \preceq \frac{n\gamma}{1-t} I.$$

**Lemma 2.** *If* $0 \preceq K - L_\gamma \preceq \frac{n\gamma}{1-t} I$ *then* $\text{bias}(L_\gamma) \leq \left(1 + \frac{\gamma/\lambda}{1-t}\right)\text{bias}(K)$.

**Lemma 3.** *If* $0 \preceq K - L_\gamma \preceq \frac{n\gamma}{1-t} I$ *and* $\lambda \geq \frac{1}{1-t}\|S\|_{op}^2 \cdot \frac{\lambda_{\max}(K)}{n}$ *then the map* $\gamma \to \text{bias}(L_\gamma)$ *is increasing. This in particular implies that under the same conditions,* $\text{bias}(L) \leq \text{bias}(L_\gamma)$.

We next prove the above lemmas.

*Proof of Lemma 1.* With $K = U\Sigma U^\top$ and $R = \Sigma^{1/2} U^\top S$, $\bar{L}_\gamma = R(R^\top R + n\gamma I)^{-1} R^\top$, we have

$$L_\gamma = U\Sigma^{1/2} \bar{L}_\gamma \Sigma^{1/2} U^\top.$$

Due to the matrix inversion lemma, we have

$$\bar{L}_\gamma = RR^\top (RR^\top + n\gamma I)^{-1}$$
$$= I - n\gamma(RR^\top + n\gamma I)^{-1}$$
$$= I - n\gamma(\Sigma + n\gamma I + RR^\top - \Sigma)^{-1}$$
$$= I - n\gamma(\Sigma + n\gamma I)^{-1/2}(I - D)^{-1}(\Sigma + n\gamma I)^{-1/2}$$

with

$$D = (\Sigma + n\gamma I)^{-1/2}(\Sigma - RR^\top)(\Sigma + n\gamma I)^{-1/2}$$
$$= \Phi - \Phi^{1/2} U^\top S S^\top U \Phi^{1/2},$$

and $\Phi = \Sigma(\Sigma + n\gamma I)^{-1}$. This shows that for any $\gamma \geq 0$

$$L_\gamma \preceq L \preceq K.$$

Now if $\lambda_{\max}(D) \leq t$ for $t \in (0,1)$,

$$I - \bar{L}_\gamma \preceq \frac{n\gamma}{1-t}(\Sigma + n\gamma I)^{-1}$$

which implies

$$0 \preceq K - L_\gamma \preceq \frac{n\gamma}{1-t}K(K + n\gamma I)^{-1} \preceq \frac{n\gamma}{1-t}I.$$

□

*Proof of Lemma 2.* This proof was communicated to us by Francis Bach [2].

Since $K - L_\gamma$ commutes with the identity, we have

$$(K - L_\gamma)^2 \preceq \frac{n^2\gamma^2}{(1-t)^2}I.$$

Now,

$$\|(L_\gamma + n\lambda I)^{-1}f^* - (K + n\lambda I)^{-1}f^*\|_2 = \|(L_\gamma + n\lambda I)^{-1}(K - L_\gamma)(K + n\lambda I)^{-1}f^*\|_2$$
$$\leq \|(L_\gamma + n\lambda I)^{-1}(K - L_\gamma)\|_{\mathrm{op}} \cdot \|(K + n\lambda I)^{-1}f^*\|_2.$$

On the other hand,

$$\|(L_\gamma + n\lambda I)^{-1}(K - L_\gamma)\|_{\mathrm{op}}^2 = \|(L_\gamma + n\lambda I)^{-1}(K - L_\gamma)^2(L_\gamma + n\lambda I)^{-1}\|_{\mathrm{op}}$$
$$\leq \frac{n^2\gamma^2}{(1-t)^2}\|(L_\gamma + n\lambda I)^{-2}\|_{\mathrm{op}}$$
$$\leq \frac{n^2\gamma^2}{(1-t)^2}\|(L_\gamma + n\lambda I)^{-1}\|_{\mathrm{op}}^2.$$

This yields,

$$\|(L_\gamma + n\lambda I)^{-1}f^*\|_2 \leq \|(K + n\lambda I)^{-1}f^*\|_2 + \|(L_\gamma + n\lambda I)^{-1}f^* - (K + n\lambda I)^{-1}f^*\|_2$$
$$\leq \|(K + n\lambda I)^{-1}f^*\|_2 \cdot \left(1 + \frac{n\gamma}{1-t}\|(L_\gamma + n\lambda I)^{-1}\|_{\mathrm{op}}\right)$$
$$\leq \|(K + n\lambda I)^{-1}f^*\|_2 \cdot \left(1 + \frac{\gamma/\lambda}{1-t}\right).$$

Hence we have the bias inequality

$$\mathrm{bias}(L_\gamma) \leq \left(1 + \frac{\gamma/\lambda}{1-t}\right)\mathrm{bias}(K).$$

□

*Proof of Lemma 3.* Let $\varphi(\gamma) = f^{*\top}(L_\gamma + n\lambda I)^{-2}f^*$. The task is to prove that $\varphi$ is increasing if $\lambda \geq \frac{1}{1-t}\|S\|_{\mathrm{op}}^2 \frac{\lambda_{\max}(K)}{n}$. We do so by computing the derivative of $\varphi$ and showing that $\varphi' \geq 0$. Let $\gamma, \gamma' > 0$. We have

$$\varphi(\gamma) - \varphi(\gamma') = f^{*\top}\left((L_\gamma + n\lambda I)^{-2} - (L_{\gamma'} + n\lambda I)^{-2}\right)f^*$$
$$= f^{*\top}(L_\gamma + n\lambda I)^{-2}\left((L_{\gamma'} + n\lambda I)^2 - (L_\gamma + n\lambda I)^2\right)(L_{\gamma'} + n\lambda I)^{-2}f^*$$
$$= f^{*\top}(L_\gamma + n\lambda I)^{-2}\left((L_{\gamma'}^2 - L_\gamma^2) + 2n\lambda(L_{\gamma'} - L_\gamma)\right)(L_{\gamma'} + n\lambda I)^{-2}f^*.$$

Now we compute the terms $L_{\gamma'} - L_\gamma$ and $L_{\gamma'}^2 - L_\gamma^2$:

$$L_{\gamma'} - L_\gamma = KS(S^\top KS + n\gamma'I)^{-1}S^\top K - KS(S^\top KS + n\gamma I)^{-1}S^\top K$$
$$= KS(S^\top KS + n\gamma'I)^{-1}\left(n(\gamma - \gamma')\right)(S^\top KS + n\gamma I)^{-1}S^\top K.$$

And

$$L_{\gamma'}^2 - L_{\gamma}^2 = KS(S^\top KS + n\gamma'I)^{-1}S^\top K^2 S(S^\top KS + n\gamma'I)^{-1}S^\top K$$
$$- KS(S^\top KS + n\gamma I)^{-1}S^\top K^2 S(S^\top KS + n\gamma I)^{-1}S^\top K$$
$$= KS(S^\top KS + n\gamma'I)^{-1}S^\top K^2 S(S^\top KS + n\gamma'I)^{-1}S^\top K$$
$$- KS(S^\top KS + n\gamma'I)^{-1}S^\top K^2 S(S^\top KS + n\gamma I)^{-1}S^\top K$$
$$+ KS(S^\top KS + n\gamma'I)^{-1}S^\top K^2 S(S^\top KS + n\gamma I)^{-1}S^\top K$$
$$- KS(S^\top KS + n\gamma I)^{-1}S^\top K^2 S(S^\top KS + n\gamma I)^{-1}S^\top K$$
$$= KS(S^\top KS + n\gamma'I)^{-1}S^\top K^2 S \left[(S^\top KS + n\gamma'I)^{-1} - (S^\top KS + n\gamma I)^{-1}\right] S^\top K$$
$$+ KS \left[(S^\top KS + n\gamma'I)^{-1} - (S^\top KS + n\gamma I)^{-1}\right] S^\top K^2 S(S^\top KS + n\gamma'I)^{-1}S^\top K.$$

The first term is the last equality above is equal to

$$n(\gamma - \gamma') \cdot KS(S^\top KS + n\gamma'I)^{-1}S^\top K^2 S(S^\top KS + n\gamma'I)^{-1}(S^\top KS + n\gamma I)^{-1}S^\top K,$$

and the second one is equal to

$$n(\gamma - \gamma') \cdot KS(S^\top KS + n\gamma'I)^{-1}(S^\top KS + n\gamma I)^{-1}S^\top K^2 S(S^\top KS + n\gamma'I)^{-1}S^\top K.$$

Now combining the above and taking the limit $\gamma' \to \gamma$ we have

$$\lim_{\gamma' \to \gamma} \frac{\varphi(\gamma) - \varphi(\gamma')}{n(\gamma - \gamma')} =$$
$$f^{*\top}(L_\gamma + n\lambda I)^{-2}KS(S^\top KS + n\gamma I)^{-1} \cdot Q \cdot (S^\top KS + n\gamma I)^{-1}S^\top K(L_\gamma + n\lambda I)^{-2}f^*,$$

with

$$Q = 2n\lambda I + S^\top K^2 S(S^\top KS + n\gamma I)^{-1} + (S^\top KS + n\gamma I)^{-1}S^\top K^2 S := 2n\lambda I + \bar{Q}.$$

Therefore, the function $\varphi$ is increasing for all $\gamma$ such that $Q \succeq 0$, and the latter is true if $2n\lambda \geq -\lambda_{\min}(\bar{Q})$. Moreover, since $\bar{Q}$ is symmetric we have

$$\lambda_{\min}(\bar{Q}) \geq -\|\bar{Q}\|_{\text{op}} \geq -2\|S^\top K^2 S(S^\top KS + n\gamma I)^{-1}\|_{\text{op}},$$

and it is sufficient to verify the condition

$$n\lambda \geq \|S^\top K^2 S(S^\top KS + n\gamma I)^{-1}\|_{\text{op}}. \tag{2}$$

Now we finish the proof by showing that the above operator norm is smaller than $\frac{1}{1-t}\|S\|_{\text{op}}^2\lambda_{\max}(K)$. We have

$$n\gamma S^\top K^2 S(S^\top KS + n\gamma I)^{-1} = S^\top K^2 S(S^\top KS + n\gamma I)^{-1}(n\gamma I + S^\top KS - S^\top KS)$$
$$= S^\top K^2 S - S^\top K^2 S(S^\top KS + n\gamma I)^{-1}S^\top KS$$
$$= S^\top K(K - KS(S^\top KS + n\gamma I)^{-1}S^\top K)S$$
$$= S^\top K(K - L_\gamma)S.$$

Taking operator norms, and using the assumption $0 \preceq K - L_\gamma \preceq \frac{n\gamma}{1-t}I$,

$$n\gamma\|S^\top K^2 S(S^\top KS + n\gamma I)^{-1}\|_{\text{op}} \leq \|S^\top\|_{\text{op}} \|K\|_{\text{op}} \frac{n\gamma}{1-t} \|S\|_{\text{op}}.$$

Hence, (2) is satisfied if $n\lambda \geq \frac{1}{1-t}\|S\|_{\text{op}}^2\lambda_{\max}(K)$ therefore concluding the proof. $\square$

## B   Proof of theorem 2

The proof uses the matrix Bernstein inequality (see e.g. Theorem 6.1.1 in [3]):

**Theorem 1.** *Consider a sequence $(X_k)$ of independent random symmetric matrices with dimension $d$. Assume that $\mathsf{E}(X_k) = 0$, $\lambda_{\max}(X_k) \leq R$, and let $Y = \sum_k X_k$. Furthermore, assume that there exists $\sigma > 0$ such that $\|\mathsf{E}(Y^2)\|_{op} \leq \sigma^2$. Then*

$$\Pr\left(\lambda_{\max}(Y) \geq t\right) \leq d \exp\left(\frac{-t^2/2}{\sigma^2 + Rt/3}\right).$$

Next, we exhibit the sequence $(X_k)$ and $Y$ in our case. We have

$$\Psi\Psi^\top = \sum_{i=1}^m \psi_i \psi_i^\top$$

and

$$\Psi SS^\top \Psi^\top = \frac{1}{p} \sum_{i \in I} \frac{1}{p_i} \psi_i \psi_i^\top = \frac{1}{p} \sum_{i=1}^m \sum_{k=1}^p \frac{1}{p_i} z_{ik} \psi_i \psi_i^\top$$

where $(z_{ik})_{1 \leq i \leq m}$ are i.i.d. binary random vectors for $k \in \{1, \cdots, p\}$ with $\Pr(z_{ik} = 1) = p_i$ (i.e. $(z_{ik})_{1 \leq i \leq m}$ is the indicator of the chosen column at trial $k$). Let $Y = \Psi\Psi^\top - \Psi SS^\top \Psi^\top$, then

$$Y = \frac{1}{p} \sum_{k=1}^p \sum_{i=1}^m (1 - \frac{z_{ik}}{p_i}) \psi_i \psi_i^\top.$$

We choose $X_k$ to be $\frac{1}{p} \sum_{i=1}^m (1 - \frac{z_{ik}}{p_i}) \psi_i \psi_i^\top$ for every $k \in \{1, \cdots, p\}$. Now we verify the assumptions of the above theorem. The matrices $X_k$ inherit independence from the random vectors $(z_{ik})_{1 \leq i \leq m}$, and we have $\mathsf{E}(X_k) = 0$, and $\lambda_{\max}(X_k) \leq \frac{1}{p} \lambda_{\max}(\sum_{i=1}^m \psi_i \psi_i^\top) = \frac{1}{p} \lambda_{\max}(\Psi\Psi^\top)$. Now we control the spectral norm of the second moment of $Y$. Again with $\mathsf{E}(X_k) = 0$ we have $\mathsf{E}(Y^2) = \sum_{k,k'=1}^p \mathsf{E}(X_k X_{k'}) = \sum_{k=1}^p \mathsf{E}(X_k^2)$. And for $k \in \{1, \cdots, p\}$

$$\mathsf{E}(X_k^2) = \frac{1}{p^2} \sum_{i,i'=1}^m \mathsf{E}\left(\left(1 - \frac{z_{ik}}{p_i}\right)\left(1 - \frac{z_{i'k}}{p_{i'}}\right)\right) \psi_{i'} \psi_{i'}^\top \psi_i \psi_i^\top$$

$$= \frac{1}{p^2} \sum_{i,i'=1}^m \left(\frac{\mathsf{E}(z_{ik} z_{i'k})}{p_i p_{i'}} - 1\right) \psi_{i'} \psi_{i'}^\top \psi_i \psi_i^\top.$$

To proceed, observe that for $i \neq i'$, $z_{ik} z_{i'k} = 0$ since only one column is chosen at a time. This yields

$$\mathsf{E}(X_k^2) = \frac{1}{p^2} \sum_{i=1}^m \frac{\mathsf{E}(z_{ik}^2)}{p_i^2} \psi_i \psi_i^\top \psi_i \psi_i^\top - \frac{1}{p^2} \sum_{i,i'=1}^m \psi_{i'} \psi_{i'}^\top \psi_i \psi_i^\top$$

$$= \frac{1}{p^2} \sum_{i=1}^m \frac{1}{p_i} \|\psi_i\|_2^2 \psi_i \psi_i^\top - \left(\frac{1}{p} \sum_{i=1}^m \psi_i \psi_i^\top\right)^2$$

$$\preceq \frac{1}{p^2} \sum_{i=1}^m \frac{\|\psi_i\|_2^2}{p_i} \psi_i \psi_i^\top.$$

Given that the probability distribution $(p_i)$ verifies $p_i \geq \beta \frac{\|\psi_i\|_2^2}{\|\Psi\|_F^2}$, we get $\mathsf{E}(Y^2) \preceq \frac{\|\Psi\|_F^2}{\beta p} \sum_{i=1}^m \psi_i \psi_i^\top = \frac{\|\Psi\|_F^2}{\beta p} \Psi\Psi^\top$. Hence $\|\mathsf{E}(Y^2)\|_{op} \leq \frac{\|\Psi\|_F^2}{\beta p} \lambda_{\max}(\Psi\Psi^\top)$. We now apply the theorem with $R = \frac{1}{p} \lambda_{\max}(\Psi\Psi^\top)$ and $\sigma^2 = \frac{\|\Psi\|_F^2}{\beta p} \lambda_{\max}(\Psi\Psi^\top)$ which leads to the desired result.

## C   Proof of theorem 3

**Monotonicity of the variance.**   First of all, we observe that the variance of the estimator $\hat{f}_K$ is matrix-increasing as a function of $K$. Indeed, we have

$$\text{variance}(K) = \frac{\sigma^2}{n} \text{Tr}(K^2 (K + n\lambda I)^{-2}) = \frac{\sigma^2}{n} \sum_{j=1}^n \frac{\lambda_j(K)^2}{(\lambda_j(K) + n\lambda)^2},$$

where $\lambda_j(K)$ is the $j$th eigenvalue of $K$ arranged in a decreasing order. The function $x \to \frac{x^2}{(x+n\lambda)^2}$ is increasing for $x \geq 0$. Moreover, if $L \preceq K$ then by the Courant-Fischer minimax principle $\lambda_j(L) \leq \lambda_j(K)$ for all $j$ (e.g. see Corollary III.1.2 in [4]).

**Risk bound.** Now, using Theorem 1 combined with the above fact, we have

$$\mathsf{E}_\xi \|\hat{f}_L - f^*\|_2^2 = \text{bias}(L)^2 + \text{variance}(L)$$

$$\leq \left(1 + \frac{\gamma/\lambda}{1-t}\right)^2 \text{bias}(K)^2 + \text{variance}(K)$$

$$\leq \left(1 + \frac{\gamma/\lambda}{1-t}\right)^2 (\text{bias}(K)^2 + \text{variance}(K))$$

$$= \left(1 + \frac{\gamma/\lambda}{1-t}\right)^2 \mathsf{E}_\xi \|\hat{f}_L - f^*\|_2^2$$

We set $\gamma = \lambda\epsilon$ and $t = 1/2$. The above holds if $\lambda_{\max}\left(\Phi - \Phi^{1/2}U^\top SS^\top U\Phi^{1/2}\right) \leq t$ and $n\lambda \geq \frac{1}{1-t}\|S\|_{\text{op}}^2 \lambda_{\max}(K)$. Now let $\Psi = \Phi^{1/2}U^\top$. Then we have $\|\psi_i\|_2^2 = l_i(\gamma)$ and $\|\Psi\|_F^2 = d_{\text{eff}}$. Using Theorem 2 on $\Psi$, and given that $\lambda_{\max}(\Psi\Psi^\top) = \lambda_{\max}(\Phi) \leq 1$, for the result to hold with probability at least $1 - \rho$, it is sufficient to set $p$ such that $n \exp\left(\frac{-p(1/2)^2/2}{d_{\text{eff}}/\beta + 1/6}\right) \leq \rho$ which gives the desired lower bound $p \geq 8(d_{\text{eff}}/\beta + 1/6)\log\left(\frac{n}{\rho}\right)$.

**Remark:** Note that if one uses the regularized Nyström approximation $L_\gamma = KS(S^\top KS + n\gamma I)^{-1}S^\top K$ with $\gamma = \lambda\epsilon$ instead of $L = KS(S^\top KS)^\dagger S^\top K$ in the algorithm then the proof would now be complete and the condition condition $n\lambda \geq \frac{1}{1-t}\|S\|_{\text{op}}^2 \lambda_{\max}(K)$ is not necessary. If one uses $L$, then this latter condition needs to be verified to insure monotonicity of the bias (see Lemma 3).

**Controlling $\|S\|_{\text{op}}$.** Now it remains to control the operator norm of the sketching matrix $S$ appearing in the lower bound on $\lambda$. To this end we use a variant of the matrix Bernstein inequality (Theorem 1) for controlling operator norms of random matrices (see Corollary 6.2.1 in [3]).

**Theorem 2.** *Consider a sequence $(X_k)$ of independent random symmetric matrices with dimension $d \times d$. Assume that $\mathsf{E}(X_k) = 0$, $\|X_k\|_{\text{op}} \leq R$, and let $Y = \sum_k X_k$. Furthermore, assume that there exists $\sigma > 0$ such that $\|\mathsf{E}(Y^2)\|_{\text{op}} \leq \sigma^2$. Then*

$$\Pr\left(\|Y\|_{\text{op}} \geq t\right) \leq 2d \exp\left(\frac{-t^2/2}{\sigma^2 + Rt/3}\right).$$

We are interested in the sum

$$Y = SS^\top - I = \frac{1}{p}\sum_{k=1}^p \sum_{i=1}^n \left(\frac{z_{ik}}{p_i} - 1\right)e_i e_i^\top,$$

and similarly to the previous section we consider the sequence $X_k = \frac{1}{p}\sum_{i=1}^n (\frac{z_{ik}}{p_i} - 1)e_i e_i^\top$ where $z_{ik}$ is defined as before and $(e_i)_{1\leq i \leq n}$ in the standard basis in $\mathbb{R}^n$. Since $p_i \geq \beta \cdot l_i(\lambda\epsilon)/d_{\text{eff}}$ with $d_{\text{eff}} = \sum_{i=1}^n l_i(\lambda\epsilon)$ we have

$$\|X_k\|_{\text{op}} \leq \frac{1}{p}\max_i \left(\frac{d_{\text{eff}}}{\beta l_i(\lambda\epsilon)} - 1\right) = \frac{1}{p}\left(\frac{d_{\text{eff}}}{\beta\underline{l}} - 1\right) \leq \frac{d_{\text{eff}}}{p\beta\underline{l}},$$

with $\underline{l} = \min_i l_i(\lambda\epsilon)$. On the other hand,

$$\mathsf{E}(X_k^2) = \frac{1}{p^2}\sum_{i=1}^n \mathsf{E}\left(\left(\frac{z_{ik}}{p_i} - 1\right)^2\right)e_i e_i^\top = \frac{1}{p^2}\sum_{i=1}^n \left(\frac{1}{p_i} - 1\right)e_i e_i^\top \preceq \frac{1}{p^2}\frac{d_{\text{eff}}}{\beta\underline{l}}I.$$

Hence

$$\|\mathsf{E}(Y^2)\|_{\text{op}} \leq \frac{1}{p}\frac{d_{\text{eff}}}{\beta\underline{l}}.$$

By choosing $\sigma^2 = R = \frac{1}{p}\frac{d_{\text{eff}}}{\beta \underline{l}}$, we have $\|SS^\top - I\|_{\text{op}} \leq t$ with probability at least $1 - 2n\exp\left(-\frac{t^2/2}{R(1+t/3)}\right)$. Taking $t = \max\left\{1, \frac{8d_{\text{eff}}}{3\beta\underline{l}\cdot p}\log\left(\frac{2n}{\rho}\right)\right\}$, the latter probability is greater than $1 - \rho$, and by the triangle inequality: $\|S\|^2_{\text{op}} \leq 1 + t$ with the same probability. By taking $p \geq 8(d_{\text{eff}}/\beta + 1/6)\log\left(\frac{n}{\rho}\right)$ (thereby verifying the condition from the previous paragraph) we have

$$\frac{8d_{\text{eff}}}{3\beta\underline{l}\cdot p}\log\left(\frac{2n}{\rho}\right) \leq \frac{1}{3\underline{l}}\cdot\frac{d_{\text{eff}}}{(d_{\text{eff}}+\beta/6)}\cdot\frac{\log\left(\frac{2n}{\rho}\right)}{\log\left(\frac{n}{\rho}\right)} \leq \frac{1}{3\underline{l}}\cdot\left(1 + \frac{\log 2}{\log\left(\frac{n}{\rho}\right)}\right) \leq \frac{1}{\underline{l}}$$

if $n \geq 2$, and therefore $\|S\|^2_{\text{op}} \leq 1 + 1/\underline{l}$ (since $\underline{l} \leq 1$) with probability at least $1 - \rho$.

## D  Proof of theorem 4

First, it is clear that

$$\tilde{l}_i = e_i^\top B(B^\top B + n\lambda I)^{-1}B^\top e_i$$
$$= e_i^\top BB^\top(BB^\top + n\lambda I)^{-1}e_i$$
$$= \text{diag}(L(L + n\lambda I)^{-1})_i$$

with $e_i$ the $i$-th element of the standard basis in $\mathbb{R}^n$. Now we bound the approximations $\tilde{l}_i$ by comparing the matrices $L(L+n\lambda I)^{-1}$ and $K(K+n\lambda I)^{-1}$ with respect to the semidefinite order. Since $L \preceq K$ (Appendix A) and the map $K \to K(K + n\lambda I)^{-1}$ is matrix-increasing, we immediately get the upper bound $\tilde{l}_i \leq l_i(\lambda)$ for all $i \in \{1, \cdots, n\}$. Next we derive the lower bound. For $\gamma > 0$, we consider again the regularized approximation $L_\gamma = KS(S^\top KS + n\gamma I)^{-1}S^\top K$ with $S \in \mathbb{R}^{n\times p}$ the sketching matrix. Due the matrix inversion lemma, $L_\gamma \preceq L$ (Appendix A). Hence to get a lower bound on $\tilde{l}_i$, it suffices to obtain a lower bound for the same quantity when $L$ is replaced by $L_\gamma$. We proved in Appendix A that if

$$\lambda_{\max}\left(\Psi\Psi^\top - \Psi SS^\top\Psi^\top\right) \leq t$$

for $t \geq 0$ with $\Psi = \Phi^{1/2}U^\top$, $\Phi = \Sigma(\Sigma + n\gamma I)^{-1}$ then

$$K - L_\gamma \preceq \frac{n\gamma}{1-t}K(K + n\gamma I)^{-1} \preceq \frac{n\gamma}{1-t}I.$$

Therefore

$$L_\gamma(L_\gamma + n\lambda I)^{-1} \succeq (K - \frac{n\gamma}{1-t}I)(K + n\lambda I)^{-1}$$

$$\succeq K(K + n\lambda I)^{-1} - \frac{\gamma/\lambda}{1-t}I,$$

where the last line follows by distributing the product and using the inequality $K + n\lambda I \succeq n\lambda I$ for the second term. Hence $\tilde{l}_i \geq l_i(\lambda) - \frac{\gamma/\lambda}{1-t}$. Now we choose again $t = 1/2$ and $\gamma = \epsilon\lambda$ for $\epsilon \in (0, 1/2)$, we get the additive error bound on $\tilde{l}_i$ and similarly to the proof of Theorem 3, it suffices to have $p \geq 8(d_{\text{eff}}/\beta + 1/6)\log\left(\frac{n}{\rho}\right)$. To finish the proof, we choose the sampling distribution $(p_i)_i$ and $\beta$ appropriately. Since

$$l_i(\gamma) = \sum_{j=1}^n \frac{\sigma_j}{\sigma_j + n\gamma}U_{ij}^2 \leq \sum_{j=1}^n \frac{\sigma_j}{n\gamma}U_{ij}^2 = \frac{1}{n\gamma}K_{ii},$$

by choosing $p_i = K_{ii}/\text{Tr}(K)$, we have $p_i \geq \beta\, l_i(\lambda\epsilon)/\sum_{i=1}^n l_i(\lambda\epsilon)$ with $\beta = n\lambda\epsilon d_{\text{eff}}/\text{Tr}(K)$, which yields $d_{\text{eff}}/\beta = \text{Tr}(K)/(n\lambda\epsilon)$.

As for the multiplicative error bound, using $K - L_\gamma \preceq \frac{n\gamma}{1-t}K(K + n\gamma)^{-1}$ we get

$$L_\gamma(L_\gamma + n\lambda I)^{-1} \succeq (K - \frac{n\gamma}{1-t}K(K + n\gamma)^{-1})(K + n\lambda I)^{-1}$$

$$= K(K + n\lambda I)^{-1}(I - \frac{n\gamma}{1-t}(K + n\gamma I)^{-1}).$$

For $t = 1/2$, $I - \frac{n\gamma}{1-t}(K + n\gamma I)^{-1} = (K - n\gamma I)(K + n\gamma I)^{-1} \succeq \frac{\sigma_n - n\gamma}{\sigma_n + n\gamma}I$. The result follows.