[Reviews · NeurIPS 2015]

Submitted by Assigned_Reviewer_1

This paper studies low-rank kernel matrix approximation in the context of kernel ridge regression. The main contribution is to show that the rank of the approximation can be chosen to be linear in the effective dimensionality for statistical guarantees, improving the result in [1] where the rank is required to be linear in the degrees of freedom. The key point is using Nystrom approximation with non-uniform sampling strategy.

Some comments & suggestions: The paper is well written and structured. The reduction in sampling complexity is novel to my knowledge, but as the authors indicated, the proofs on the statistical guarantees largely follow from the work in [1], which somewhat affects the novelty of the paper. It seems that there is an error in the proof of theorem 2 (line 074 in the supplementary material). Specifically, the random variable on the left-hand side corresponds to column sampling with replacement (exactly p columns are picked out), while, for the random variable on the right-hand side, the number of picked columns takes values over the interval {0,1,...,mp}. So these two random variables are not equal in my opinion. As this equality is the foundation of the subsequent deduction, I suggest the authors to check their proof carefully (perhaps, you wanted to use z_{ik} for some different meaning: the indicator of the k-th sampling, i.e., z_{ik}=1 iff the k-th sampling selects the i-th column. However, in this case, the identity in the line 096 would not hold).

Some minor comments: (1) line 085, "which if" ---------> "whether if" (2) line 101, "l:(Y,X)"

---------> "l:(Y,Y)" (3) Eq. (4), you miss some "sigma^2" in front of \xi (4) line 216, "in conjunction will" ----------> "in conjunction with" (5) line 148 (supplementary material), "\gamma/\lambda"

-----------> "n\gamma"

Reference.

[1] Bach, Francis. "Sharp analysis of low-rank kernel matrix approximations." arXiv preprint arXiv:1208.2015 (2012).
Summary: A moderately novel paper studying low-rank kernel matrix approximations in the context of kernel ridge regression. There is an error in the proofs.

Submitted by Assigned_Reviewer_2

Quality: Paper is well written.

Clarity: The statement and the results are clear and well motivated. The proofs seem sound.

Originality: The paper is an evolution of [2]. While [2] focuses on uniform subsampling methods, this paper focuses on subsampling methods based on leverage scores. The proof techniques in this paper are a variation of the ones in [2].

Significance: With respect to [2], this paper addresses the more interesting and useful case of subsampling with leverage scores. This paper, as [2], proposes an analysis in the fixed design setting (that is remarkably different fromt the classical statistical machine learning setting that is a nontrivial extension of the random design setting), it is definitely a first step towards a theoretical analysis of such efficient approximation scheme for KRR.
Summary: The paper studies some theoretical properties of the Nystrom approximation scheme applied to Kernel Ridge Regression (KRR) in the fixed design setting. It proves that when the points in Nystrom are selected by means of leverage scores, then the approximated KRR algorithm has an accuracy that is of the same order of the standard KRR.

The algorithm has been already studied from a computational point of view for example in [1] and the proof techniques are a variation on the ones in [2]. Indeed [2] proves that in the fixed design setting Nystrom KRR approximation with uniform selection method has an accuracy that is of the same order than standard KRR.

The interesting aspect of KRR Nystrom with lev. scores is that it needs far less points than the ones required by using a uniform selection method. Indeed, as pointed out in the introduction, the proposed approximation scheme is among the fastest in literature for which there exist statistical guarantees.

Despite the fact that the statistical properties of the algorithm are studied in the fixed design setting, that is simpler and remarkably different than the classical statistical machine learning setting (random design), the proposed result can be considered a first step in analyzing the properties of a very useful approximation scheme for KRR from a theoretical viewpoint.

Submitted by Assigned_Reviewer_3

- While the paper appears to be

sound and well presented on the theory side, it seems that the authors forgot to put the MSE results in the table of accuracy and to compare it to other methods. Timing of the experiments is also not reported which makes it hard to grasp the importance of this method, and how it compares to other methods with classical leverage scores.

- More convincing experiments are needed to see how those new scores interact with the nystrom method in term of accuracy and how they compare with other scores.

- One would expect that Nystrom sampling itself is doing a regularization and hence no need for lambda, can those scores be computed without the need of lambda?

Summary: This paper presents an approach to improve the running time of kernel methods. New notion of leverage score is introduced in the paper, that reduces the complexity of the nystrom method to the effective dimension of the problem.

Submitted by Assigned_Reviewer_4

This paper extends on previous work relating to the Nystrom method of low rank approximation of SPSD (e.g. kernel) matrices, showing that tighter bounds can be derived that the sharp analysis of Bach using a non-uniform sampling strategy that attempts to control the bias. The idea of non-uniform sampling for the Nystrom approximation is not itself new, but the analysis goes further than this, and holds for any sketching matrix that meets certain conditions. This, combined with the resultant bounds, is a significant contribution.

Comments/criticisms: - the title is "Fast Randomized Kernel Methods ..." whereas currently the analysis only holds for Kernel Ridge Regression. **Note that the title was changed for final submission** - it seems that this method will not work for any normalised kernel matrix (i.e. where f(x) = \phi(x)/ \left| \phi(x) \right|), since the p_i's will be uniform. For example, the RBF kernel will always have 1's on the diagonal, so K_{ii} / TR(K) will just be 1/n? - in the first experiment, what is the MSE when using full KRR on this dataset? does the lambda ridge method converge to this error? - in the second experiment (Table 1), why haven't MSE's been reported? A comparison with existing methods should also be done here.

Specific comments: - for ref [1] it would be better to refer to the published ICML paper than the unpublished arxiv paper unless there's a good reason not to (http://machinelearning.wustl.edu/mlpapers/bibtex/icml2013_gittens13) - for ref [5] same as above (http://jmlr.org/proceedings/papers/v32/ma14.html) - L67, L256 the authors refer to Bach's d_{mof} as "maximal degrees of freedom", but I believe Bach called this "marginal degrees of freedom" - eq 4 (and later) the first term is given as bias(K)^2 whereas Bach denoted this same term as simply bias(K) - is there a reason for changing this? - L158

The operator  is used for the first time. I presume you are referring to the linear matrix inequality X \succeq Y \Longleftrightarrow \lambda_{min}(X-Y)\geq 0, but I'm not sure all readers will be aware of this - L162 derived of => derived from - Theorem 2. State what t is (i.e. t \in (0, 1)?) - L302-305 Section 3.5. The last sentence in the first paragraph is unclear and too long, please reword this - L373 experiments -> experiment - Figure 1. These figures are too small to be readable. I realise space is an issue but please make these more readable if possible. - L398 haven't we known there existence => had we not known of their existence - L399 determine \lambda => determine \lambda and \sigma - Table 1. \sigma was used earlier in the paper to refer to eigenvalues, now it refers to the RBF parameter - could a different symbol be used? - throughout the paper (and references inverted commas are the wrong way round, caused by using the double quote " character instead of two single quotes '' - In the references the capitalisation is wrong in many cases: conference/journal papers should have title case (except for proper nouns and abbreviations), whereas books should have every word capitalized. Nystrom should appear as Nystr\"om. Journal papers should have issue numbers and page numbers.
Summary: A strong paper from a theoretical perspective, that builds on previous work on the statistical properties of low rank matrix approximation, although the experiments could be stronger.

Author Feedback
Author rebuttal: On the whole, the reviewers acknowledged the high quality of our theoretical results and the importance of the subject. Four reviewers gave the paper high marks. Their comments included that our method "needs far less points than the ones required by using a uniform selection method," that our method is "among the fastest in the literature for which there exist statistical guarantees," that ours is "a strong paper from a theoretical perspective," and that it is "an extremely well written paper presenting useful and very thorough improvements approximate KRR." The other two reviews were by a reviewer who pointed out a minor easily-fixable imprecision in the statement of our theoretical result and one who provided a low-quality undetailed review. Several reviewers pointed out correctly that this is more of a theoretical paper.

Regarding comments about the empirical section:
-We decided to compare our method only to pre-existing method in the Nystrom family and not to other methods more generally. The reason was due to space and since previous literature already extensively compared Nystrom methods with other kernel approximation schemes [1,2,5].
-In experiment 1, the MSE when the full kernel is used is asymptotically approached as p gets large in all cases, albeit at different speeds, the lambda-ridge scores method being the fastest.
-The MSE was not reported in Table 1, as that compares d_{mof} and d_{eff}. The MSE is irrelevant there since only one method (sampling with lev. scores) is used.
-Our algorithm was not optimized to yield the best run time, but we exhibited the main bottlenecks that such an algorithm has to overcome, and we showed that they indeed can be overcome. Namely: (1) we bring down the number of columns sampled in order to provide a statistically accurate prediction to the effective dimension d_{eff}, and (2) we minimize the memory requirement of the computation, i.e., no need to form and store the entire kernel matrix explicitly. Both results are new for this class of sampling methods.

Reviewer 1:
The "error" the reviewer is referring to is an imprecision in our statement, and it is easy to correct by being more precise about the sampling process (i.e. definition of the variables z_{ik}). We will do so, and we are grateful for that being pointed out to us. Thanks for spotting the typos.

Reviewer 2:
The random design setting is treated in the very recent paper [Rudi et al., arXiv 1507.04717] that builds on the methods we introduce in this submission. They show that leverage-based Nystrom has the same desirable risk properties, w.h.p. on the randomness of the data.

Reviewer 3:
-The title is intended to emphasize "fast" and "statistical guarantees" in the context of kernel methods, not that we claim full generality on all kernel methods. This being said, and while we prove our results in the context of the \ell_2 loss, there is evidence that \ell_2-like properties in statistical estimation extend to losses that verify a self-concordance property. E.g., the logistic loss, where a bias-variance decomposition holds asymptotically (See [Bach. Self-concordant analysis for logistic regression. EJS. 2010]). On the other hand, this problem it is still completely open for non smooth objectives (e.g., SVM).
-For the RBF case, the fact that the kernel matrix contains 1's on the diagonal does not mean the leverage scores are uniform. But in this case the approximation scheme becomes particularly simple: Theorem 4 tells you to sample uniformly O(1/\lambda) columns to get an approximation of the leverage scores.
Thanks for the minor comments.

Reviewer 5:
Nystrom is doing regularization, and this is studied in detail in a very recent paper [Rudi et al., arXiv 1507.04717] which builds on our work and others. But lambda plays other crucial roles than "just" statistical regularization. The problem stops being interesting for lambda = 0, since the effective dimension reduces to the rank of K (which is infinite at the population level) and the lambda-scores become the usual scores. Even Rudi et al. consider lambda >0.

Reviewer 6:
-The experiments show: (1) fewer columns are needed when sampling using lev. scores, as measured by the *MSE* (not by some irrelevant matrix norm), and (2) the effective dimension is often much smaller than previously defined notions of dimension.
-Line 398 is a typo. Thanks for spotting that.